# Contemporary Assessment of Post-Operative Pancreatic Fistula After Pancreatoduodenectomy in a European Hepato-Pancreato-Biliary Center: A 5-Year Experience

**DOI:** 10.3390/medicina62010094

**Published:** 2026-01-01

**Authors:** Dimitrios Vouros, Maximos Frountzas, Angeliki Arapaki, Konstantinos Bramis, Nikolaos Alexakis, Ajith K. Siriwardena, Georgios K. Zografos, Manousos Konstadoulakis, Konstantinos G. Toutouzas

**Affiliations:** 1First Propaedeutic Department of Surgery, ‘’Hippocrateion’’ General Hospital, School of Medicine, National and Kapodistrian University of Athens, 11527 Athens, Greece; froumax@hotmail.com (M.F.); aarapaki@gmail.com (A.A.); kbramis@gmail.com (K.B.); nick.alexak@gmail.com (N.A.); surg-clinic-uoa@hippocratio.gr (G.K.Z.); konstadoulakismm@yahoo.com (M.K.); tousur@hotmail.com (K.G.T.); 2Regional Hepato-Pancreato-Biliary Unit, Manchester Royal Infirmary, Manchester M13 9WL, UK; a.siriwardena@btinternet.com; 32nd Department of Surgery, Aretaieio Hospital, School of Medicine, National and Kapodistrian University of Athens, 11528 Athens, Greece

**Keywords:** pancreatoduodenectomy, pancreatic fistula, pancreatic cancer

## Abstract

*Background and Objectives*: Pancreatoduodenectomy (PD) is the primary treatment for patients with resectable, non-metastatic pancreatic adenocarcinoma and periampullary tumors. Although surgical methods and perioperative management have improved, the procedure still carries a high risk of complications, with postoperative pancreatic fistula (POPF) being the most significant. This study focuses on identifying current risk factors for POPF after PD in a single HPB center. *Materials and Methods*: We retrospectively analyzed prospectively collected data from patients undergoing PD in our department between October 2018 and April 2024. Data included demographics, comorbidities, lifestyle factors, preoperative tests (bilirubin, CA19-9, HbA1c), intraoperative variables (pancreatic texture, duct diameter), and postoperative outcomes. POPF was classified using the International Study Group of Pancreatic Surgery (ISGPS) criteria. Univariate and multivariate logistic regression analyses were performed. *Results*: A total of 118 patients underwent PD (82 males, 36 females; mean age 67 (45–85) years; mean body mass index (BMI) 26.6 kg/m^2^). POPF occurred in 37 patients (31%), with 27 Grade B (23%) and 10 Grade C (9%). The 30- and 90-day mortality rates were 5% and 12.7%, respectively. Univariate analysis showed associations between POPF and soft pancreas (*p* = 0.018), c-reactive protein (CRP) on postoperative day (POD) 5 (*p* = 0.004), and serum amylase on POD 0 (*p* = 0.008). Diabetes mellitus was associated with a lower incidence of POPF (*p* = 0.014). Multivariate analysis confirmed CRP on POD 5 (OR 1.007, *p* = 0.025) and DM (OR 0.254, *p* = 0.015), as independent factors. ROC analysis identified POD 0 amylase >113.5 U/L (AUC 0.717) and POD 5 CRP >125.3 mg/dL (AUC 0.669) as predictive values. *Conclusions*: POPF remains an important complication after PD. CRP > 126 mg/dL on POD 5 was associated with POPF and may serve as an adjunctive signal to guide further assessment, including imaging. The observed inverse association with diabetes mellitus is hypothesis-generating and should be interpreted cautiously, considering potential confounding and the influence of center volume, surgeon heterogeneity, and institutional protocols.

## 1. Introduction

Pancreatoduodenectomy (PD) is currently the only treatment with curative intent for malignancies involving the head of the pancreas [1]. Despite reductions in PD-related mortality to about 2%, postoperative morbidity remains high, ranging from 30% to 50%, with postoperative pancreatic fistula (POPF) and delayed gastric emptying (DGE) being the most common complications [2,3]. The incidence of POPF varies widely, with rates ranging from about 10% to 28% [2,3]. POPF is a common and dangerous complication that may lead to sepsis or bleeding, which in some cases may require reoperation [4]. Numerous studies have attempted to identify risk factors for clinically relevant POPF (CR-POPF) [5]. Although many studies have identified risk factors for POPF after PD, there is still no universal consensus on which factors are the most reliable, except the pancreas texture and the size of the pancreatic duct [6]. Predictive models vary widely, and some include complex or subjective variables [7]. In particular, the role of serum markers like C-reactive protein (CRP) and amylase, as well as comorbidities such as diabetes mellitus (DM), remains unclear, with conflicting results across the literature [8,9]. This study aims to retrospectively evaluate all patients who underwent PD in a single hepatopancreatobiliary (HPB) center over the past five years, with a focus on identifying reliable predictors for POPF.

## 2. Materials and Methods

### 2.1. Patients

Patients referred to our department with pancreatic or periampullary neoplasms were enrolled in our study. The unit is a tertiary referral center for a metropolitan area of 1.5 million people. The inclusion period was from October 2018 until April 2024. All patients were discussed in the local multidisciplinary team (MDT) meeting prior to any intervention. Patients who were not eligible for pancreatectomy (due to advanced disease, poor performance status, or refusal of surgery) were excluded from analysis.

### 2.2. Data Collection

Detailed clinical information was gathered for each patient, including demographic variables such as age, sex, body mass index (BMI), tobacco use, and alcohol intake. Medical background data included history of endoscopic retrograde cholangiopancreatography (ERCP), chronic pancreatitis, prior cholecystectomy, neoadjuvant chemotherapy, DM, anorexia, unintentional weight loss, and the presence of jaundice. Preoperative evaluation involved laboratory investigations measuring bilirubin, albumin, ferritin, carbohydrate antigen 19-9 (CA19-9), carcinoembryonic antigen (CEA), platelet count (PLT), and glycated hemoglobin (HbA1c). The recorded intraoperative parameters comprised pancreatic tissue consistency, diameter of the pancreatic duct, use of blood transfusions, total volume of intravenous fluids administered, size of the pancreatic transection surface, preservation or not of the pylorus, type of anastomosis performed, and total operative time. Specifically, the texture of the pancreas was assessed by the surgical team after the specimen was excised and before performing the pancreatojejunal anastomosis. The texture was classified as soft, intermediate, or hard. In some cases, preoperative computed tomography (CT) scans described pancreatic fibrosis or the presence of peripancreatic fatty tissue; however, this evaluation was not performed in all of our patients. Post-surgical outcomes included hospital length of stay (LOS), pathological findings, extent of lymph node dissection, tumor node metastasis (TNM) classification of the tumors, and whether somatostatin analogs were administered. Post-operative complications were monitored using laboratory markers, including white blood cell (WBC) count, CRP, PLT, and drain fluid amylase concentrations measured on POD 1, 3, and 5.

### 2.3. Definitions Used in This Study

Complications were classified using the Clavien–Dindo (CD) classification [10]. Major complications were defined as those necessitating general anesthesia, intensive care unit (ICU) admission, or resulting in patient mortality (grades 3B–5). POPF was defined and graded according to the updated ISGPS guidelines [11]. A biochemical leak (BL) is considered clinically insignificant, as it does not alter the standard postoperative course or extend the typical LOS. Grade B pancreatic fistula is characterized by elevated amylase levels in drain fluid accompanied by clinically significant symptoms requiring intervention. Grade C fistulas represent a more severe progression, involving organ dysfunction or critical deterioration that may necessitate reoperation. All postoperative pancreatic fistulas were retrospectively reclassified according to the 2016 ISGPS update. Biochemical leaks were analyzed separately and excluded from the definition of CR POPF (Grades B and C) used in the regression models. POPF in the results and tables refers to Grade B and C only. DM was defined and categorized based on the American Diabetes Association Professional Practice Committee definition [12]. HbA1c was measured in all patients, and those with values ≥6.5% were categorized as having newly onset DM if there was no previous history of classic symptoms of hyperglycemia or a previous diagnosis. In patients with known DM, HbA1c was also measured to assess whether their DM was well controlled. Finally, unintentional weight loss was defined as ≥5 kg or ≥5% of body weight within 6 months before surgery and analyzed as a binary categorical variable (present/absent).

### 2.4. Details of Surgical Procedures

PD involved resection of the pancreatic head, duodenum, gastric antrum (for classic Whipple), proximal jejunum, gallbladder (if not previously removed), and common bile duct. Decisions around the type of pancreatojejunal anastomosis performed were based on the surgical team’s preference and the patient’s anatomical factors, such as the diameter of the pancreatic duct and the texture of the pancreas. A plastic stent was inserted in the pancreatojejunal anastomosis, with external wirsungostomy performed in some cases. Closed vacuum drains were positioned near the pancreatojejunal and hepaticojejunal anastomoses. If two drains were placed, these were marked as the “Right” and the “Left” drain. Postoperative administration of somatostatin analogs was determined by the surgeon’s preference.

### 2.5. Statistical Analysis

The primary objective of the statistical analysis was to identify preoperative and intraoperative predictors of POPF. Postoperative parameters such as length of stay and survival were analyzed descriptively as outcomes and were not entered into the predictive models.

Power analysis [13] determined that, to detect significant differences in the proportion of patients with POPF exceeding 15%, a minimum of 89 patients would be required. Adjusting for potential losses, a 30% increase was applied, leading to a final sample size of 118 patients. These cases were prospectively collected and the analysis was performed retrospectively.

All continuous variables were tested for normal distribution using the Shapiro–Wilk statistic. As all variables had a normal distribution based on the Kolmogorov–Smirnov test, typical tests were used for the basic analysis (Student’s t-test after testing for equality of variances). Categorical variables are shown as absolute (N) and relative (%) frequencies while continuous variables are described as means and standard deviations (SDs). Pearson’s χ^2^ and Fisher’s exact statistics were used to test for associations between categorical variables.

After the results of the univariate analysis, ROC analysis was performed for each significantly associated continuous variable in order to calculate cut-off values having the higher possible sensitivity and specificity when predicting POPF. Univariate logistic regression was initially performed to evaluate the relationship between each variable and POPF. Variables with *p* < 0.10 and those considered clinically relevant were added into a multivariate model using a backward stepwise elimination approach (entry criterion *p* < 0.05, removal criterion *p* > 0.10). Model calibration was tested using the Hosmer–Lemeshow goodness-of-fit test, and model discrimination was assessed via the area under the receiver operating characteristic curve (AUC). An events-per-variable ratio above 10 was maintained to minimize overfitting. Multiple logistic regression followed, taking into consideration all variables used in the univariate analysis. After multiple testing for goodness of fit (likelihood ratio test), information criteria (Akaike information criterion (AIC)), and predictive accuracy (ROC curve) a best possible model was produced. All tests were two-sided. Due to multiple testing, Bonferroni adjustment was omitted. *p*-values were considered significant if *p* < 0.05. STATA^®^ v.18.0 (StataCorp, College Station, TX, USA) statistical software was used for the analysis.

## 3. Results

During the inclusion period, 118 patients underwent PD in our department. Eighty-two of the patients were male (70%), and thirty-six were female (30%). Patient demographics and preoperative biochemical markers are presented in Table 1. Details of the patients’ medical history and presenting symptoms are shown in Table 2. All intraoperative parameters are presented in Table 3 and Table 4.

### 3.1. Postoperative Outcome

POPF was present in 37 patients (31%), with 27 patients developing Grade B POPF (23%) and 10 patients (8%) developing Grade C (Table 5). A biochemical leak was present in 32 patients (27%). As expected, patients who developed POPF experienced more major complications (37.8% vs. 6.1%, *p* = 0.033), had a higher reoperation rate (37.8% vs. 3.7%, *p* < 0.001), and a significantly longer postoperative hospital stay (median 26 vs. 12 days, *p* < 0.001). These parameters were treated as outcomes rather than predictors in the revised analysis. The 30-day and 90-day mortality rates were 5% and 12.7%, respectively. The presence of POPF did not affect 30-day mortality (8.1% vs. 3.7%, *p* = 0.376), although the 90-day mortality rate was statistically significant between the two groups (21.6% vs. 8.6%, *p* = 0.05).

### 3.2. Histopathology

The histology reports of the patients undergoing PD is shown in Table 6. Pancreatic ductal adenocarcinoma (PDAC) was the most common pathology (83 patients, 70%). Patients with PDAC had a lower observed rate of POPF when compared with the control group (76.5% vs. 56.7%, *p* = 0.049).

### 3.3. Biochemical Markers and POPF

WBC count, CRP, PLT count, and drain amylase levels from both drains were recorded for all patients on POD 1, 3, and 5 (Table 7). Patients with POPF presented with higher CRP levels on days 1, 3, and 5, with all differences being statistically significant [131.3 vs. 109.9 mg/L (*p* = 0.018), 221.8 vs. 185.7 mg/L (*p* = 0.009), and 171.9 vs. 121.8 mg/L (*p* = 0.003), respectively]. Drain amylase levels were found to be statistically significant on POD 1 and POD 5 [2025.5 vs. 591 U/L from the right drain, *p* = 0.028; 3950.8 vs. 774.3 U/L from the left drain, *p* = 0.016, respectively]. Serum amylase on day 0 was also significantly different between the POPF and control groups, being elevated in patients who went on to develop fistula (269 vs. 160.6 U/L, *p* = 0.005).

In ROC analysis, baseline serum amylase levels above 113.5 U/L and CRP levels on POD 5 exceeding 125.3 mg/dL are both associated with the presence of POPF. The amylase threshold has a sensitivity of 77.1% and a specificity of 60%, with an area under the curve (AUC) of 0.717. The CRP threshold has a sensitivity of 70.3% and a specificity of 61.3%, with an AUC of 0.669 (Figure 1).

### 3.4. Univariate Analysis for POPF

The risk factors for the development of POPF are presented in Table 8. Patients with a soft pancreas texture were more likely to develop a POPF. The incidence of POPF in patients with soft pancreas texture was 22/51 (43.1%) and that was 15/67 (22.3%) of the patients with non-soft pancreas texture (*p* = 0.018). The presence of DM was found to be a protective factor for development of POPF. The incidence of POPF in patients without DM was 31/79 (39.2%) and that was 6/38 (15.7%) of the patients with presence of DM (*p* = 0.014). Other factors related with presence of POPF was CRP on POD 5 (OR 1.007, 1.002–1.011 95% C.I. *p* = 0.004), serum amylase on POD 0 (OR 1.003, 1.000–1.005 95% C.I. *p* = 0.008) and soft pancreas (OR 0.380, 95% C.I. 0.171–0.845, *p* = 0.018). Univariate analysis demonstrated no significant relationship between POPF and the following factors: Age, sex, horizontal and vertical pancreatic cutting surface diameter, histology, pancreatic cutting surface area, positive lymph nodes, Ca19-9, history of smoking, weight loss and duration of surgery. Pancreatic duct diameter was at the limit of a significantly statistical difference (*p* = 0.056).

### 3.5. Multivariate Analysis for POPF

The final multivariate model retained four variables: CRP on POD 5, diabetes mellitus, serum amylase on POD 0 and pancreas texture. The model demonstrated good calibration (Hosmer–Lemeshow χ^2^ = 5.55, *p* = 0.136) and satisfactory discrimination (AUC = 0.723).

In multivariate logistic regression analysis, POPF was associated with the following factors: CRP levels on POD 5 (OR 1.007, 95% CI 1.001–1.012, *p* = 0.025) and DM (OR 0.254, 95% CI 0.084–0.763, *p* = 0.015) (Table 9). POPF was not associated with pancreas texture (OR 1.747, 95% CI 0.708–4.311, *p* = 0.226) and serum amylase on POD (OR 1.002, 95% CI 0.999–1.004, *p* = 0.149).

## 4. Discussion

POPF is a critical determinant of morbidity and mortality following pancreatic surgery [15]. Various risk factors for POPF have been extensively studied. Patient-related risk factors include male sex [3,5,15], high BMI [3,5], intra-abdominal fat thickness, and preoperative nutritional optimization [5]. Pancreas-related risk factors include soft pancreatic texture [2,5,15,16], pancreatic duct diameter <3 mm [3,5,15], high-risk etiology (e.g., benign pathology or extra-pancreatic tumors) [3,5], ongoing pancreatitis, and high acinar cell density [5]. Procedure-related risk factors include intraoperative blood loss [3,5,15], intraoperative blood transfusion [3,15], and the anastomotic technique employed [2,5].

Multiple techniques in anastomosing the pancreas to the jejunum after PD have been described in the literature [17]. In our series, most of our performed anastomoses were the classic duct to mucosa, a technique initially described by Cattel and Warren in 1956, and Blumgart anastomosis [18]. In our analysis, Blumgart anastomosis was related with a lower incidence of clinically relevant pancreatic fistula (49.4 vs. 24.3%, *p* = 0.001), although this difference was not found to be statistically significant in the multivariate analysis. Blumgart anastomosis can be applicable to all patients in whom the pancreatic duct can be identified, and it is associated with lower rates of significant postoperative morbidity and mortality [19]. The technique used for the anastomosis was based on intraoperative findings such as the texture of the pancreas, the pancreatic duct diameter, and the surgeon’s preference.

DM is a known risk factor for morbidity, LOS, and mortality after surgery [20]. DM has a high prevalence (40%) in pancreatic cancer and is frequently of new onset (NODM) [21]. In studies involving patients undergoing pancreatectomy, the percentage of patients with DM is lower, ranging from 19.9% to 21.9%, with NODM ranging from 30% to 51.6% [20,22]. The relationship between DM and postoperative outcomes following PD, particularly regarding the development of POPF and long-term survival, remains uncertain [20]. Some studies have reported no significant difference in POPF incidence between diabetic and non-diabetic patients; however, they have noted reduced 3- and 5-year survival rates among those with DM [20]. In contrast, other studies have reported an inverse association, whereby diabetes has been associated with a lower incidence of POPF, with CR-POPF occurring more frequently in non-diabetic patients [22,23].

In our study, 38 patients (32.5%) were diagnosed with DM, either as NODM or as a chronic disease based on their past medical history. Interestingly, 32 out of 48 patients with no signs of POPF were diabetic (40%), compared with 6 out of 31 patients with POPF (16.2%). The difference was statistically significant (*p* = 0.011). Additionally, HbA1c values were also significantly different between the two groups (6.3% vs. 5.9%, *p* = 0.037). In both univariate and multivariate analyses, DM was not a risk factor for POPF in patients undergoing PD. Accordingly, the absence of DM emerged as being statistically associated with a higher likelihood of POPF occurrence.

The apparent inverse association between DM status and POPF observed in our cohort should be interpreted as hypothesis-generating. This association may reflect underlying pancreatic fibrosis, duct size, or other unmeasured factors, rather than a direct protective effect of DM. The limited sample size precludes definitive conclusions, and further multicentric validation is warranted.

DM does not seem to be a risk factor for POPF, as reported in a meta-analysis by Xia et al. [24]. One possible explanation for this observed association is that patients with DM are less likely to present with high-risk pancreatic features, such as soft texture and/or small duct size [22]. Additionally, patients with DM often exhibit greater pancreatic fibrosis and less fatty infiltration, factors that may correlate with a more secure pancreatojejunostomy [23]. This contrasts with non-diabetic patients, who tend to have softer pancreatic tissue with higher fat content [24].

Preoperative weight loss was observed in 84.6% of patients, with a mean reduction of 6.9 kg across the cohort. Patients without POPF reported weight loss more frequently (90%) compared with those who developed POPF (73%), a statistically significant difference (*p* = 0.018). Although the mean amount of weight lost was higher in the non-POPF group (7.7 vs. 5.3 kg), this difference did not reach statistical significance (*p* = 0.058).

Inflammatory markers such as WBC count, CRP, and procalcitonin (PCT) have been widely used as associative indicators for the presence of postoperative complications [25,26]. Specifically, the role of CRP has been extensively evaluated as a biomarker for the early identification of infectious complications following major abdominal surgeries, as well as for the detection of anastomotic leaks in colorectal surgery [27,28]. CRP is the first acute-phase reactant synthesized in the liver, with a half-life of 19 hours [26]. After stimulation, CRP levels rise above normal within 6 hours and peak at 48 hours. Normally, CRP levels gradually decline after surgery on POD 2 and 3 in the absence of inflammatory stimuli [26]. CRP has been reported as being associated with CR POPF [29]; hence, it could also have potential value as part of the early risk stratification and clinical assessment of CR POPF after PD [26,29].

In our study, mean CRP levels on POD 1, POD 3, and POD 5 were 116.5 mg/L, 197.4 mg/L, and 137.6 mg/L, respectively. CRP levels on POD 1, 3, and 5 were statistically significantly different between the two groups. In univariate logistic regression analysis, CRP on POD 5 was statistically different between the POPF and control groups (121.8 mg/L vs. 171.9 mg/L, OR 1.007, 95% CI 1.002–1.012, *p* = 0.004). CRP on POD 5 was higher in the POPF group, and this difference remained statistically significant as an association in the multivariate analysis (OR 1.008, 95% CI 1.002–1.014, *p* = 0.006). ROC analysis showed that CRP ≥ 126 mg/L on POD 5 was associated with the presence of clinically relevant pancreatic fistula, with a sensitivity and specificity of 70.27% and 61.25%, respectively (AUC: 0.669). While CRP and serum amylase demonstrated statistically significant associations with POPF, their AUC values indicate only moderate discriminative performance. These ROC-derived thresholds should therefore be interpreted as adjunctive decision-support signals rather than stand-alone triggers for imaging or intervention. Therefore, these biomarkers should not be used in isolation to establish the diagnosis of CR POPF but can serve as adjunctive decision-support signals prompting further investigation, particularly when combined with other biochemical markers and clinical assessment. For example, WBC count and PLT count were not found to be significantly associated with the presence of POPF on POD 1, 3, and 5.

Based on our study findings, we routinely measure drain amylase on postoperative day (POD) 3, as it is one of the criteria defined by the ISGPS for identifying patients with pancreatic leakage (either biochemical leak or clinically relevant POPF). Serum amylase on POD 0 may represent a useful biochemical marker for the early identification of patients at increased risk of developing CR-POPF; however, although POD 0 serum amylase did not retain statistical significance in the multivariate model, it showed moderate discriminative ability in univariate and ROC analyses (cut-off >113.5 U/L; AUC = 0.717). This finding may indicate that early postoperative hyperamylasemia is associated with pancreatic anastomotic stress or subclinical leakage before other inflammatory markers rise. On POD 5, we measure serum CRP, and if the value exceeds 126 mg/L, we consider this finding in conjunction with the overall clinical picture when deciding whether to perform a CT scan to facilitate the early recognition of local or systemic complications potentially related to POPF. If the patient appears clinically unwell or there is another indication for imaging, a CT scan is performed regardless of the CRP level on that day.

PD is recognized as one of the most complex abdominal surgical procedures, associated with a substantial risk of postoperative complications. [25]. Advances in gastrointestinal endoscopy and interventional radiology have made non operative management possible in many cases [30]. Nevertheless, surgery is still required when these measures fail or when rapid clinical deterioration dictates a definitive solution [30]. The reoperation rate in our study was 14.5%, as 17 patients underwent laparotomy for postoperative complications. The majority of patients had POPF (14 patients). The presence of CR POPF was statistically associated with reoperation in our cohort.

While our findings corroborate previously reported POPF risk factors, their validation in a single center demonstrates continued relevance without claiming external confirmation. Notably, some risk factors, such as pancreatic texture, showed strong univariate associations but lost statistical significance in multivariate modeling, highlighting a discrepancy with established risk scores and suggesting that our data may partly replicate or refine, rather than confirm, prior models. The study’s findings may also be influenced by center volume, surgeon heterogeneity, and institutional protocols, which could affect observed associations and limit external generalizability. Importantly, these results have influenced our local postoperative protocol: patients with a soft pancreatic gland, a small pancreatic duct diameter, and/or elevated CRP levels on postoperative day 5 are now managed under an intensified monitoring pathway that includes early imaging with CT, selective use of somatostatin analogs in the early postoperative period, and tailored drain management. Moreover, early multidisciplinary review involving dietitians, infectious disease specialists, and interventional radiologists is implemented for patients at higher risk of developing CR POPF.

Although our study was conducted in a single European HPB center, the variables identified—such as CRP and serum amylase—are objective, widely available, and globally applicable markers. However, differences in patient characteristics, pancreatic texture distribution, and surgical practices between institutions may affect the predictive performance of these markers. Therefore, multicenter prospective validation across diverse healthcare settings is warranted to establish global applicability.

One of the main strengths of this study is the use of a prospectively collected dataset, which allowed for detailed information to be gathered over five years. Performing the study at a single HPB unit helped ensure that surgical techniques and postoperative care were consistent, reducing variability in the results. However, as the analysis is retrospective, it carries some limitations like potential selection bias. The fact that the analysis was undertaken at a single center might also limit how applicable the findings are to other settings. Lastly, while the sample size is adequate, analyzing many variables raises the chance of type I errors.

Our center performs approximately 25 PDs annually, classifying it as a medium-volume HPB unit. This may partly explain the relatively high POPF and mortality rates observed compared with large tertiary referral centers. Furthermore, four consultant surgeons performed PDs during the study period; while no significant association was found between individual surgeon volume and POPF severity, the small sample size precluded definitive conclusions.

Although only 37 patients developed POPF, the final multivariate model included two independent variables, corresponding to an events-per-variable ratio of approximately 18. This exceeds the conventional threshold of 10 events per variable that is considered adequate for logistic regression modeling [31,32]. Nevertheless, the limited number of events and the single-center nature of the study may reduce generalizability, and future multicenter studies with larger cohorts are needed to externally validate these findings.

A final limitation of the study is that it did not include a defined follow-up endpoint, as long-term survival was not a study objective. Future research could integrate postoperative outcomes and survival endpoints in prospective analyses to assess the long-term impact of POPF.

## 5. Conclusions

In our study, CRP on POD 5 was found to be associated with the presence of CR POPF, as a CRP level ≥126 mg/dL showed a sensitivity of 70.27% and a specificity of 61.25%, with an AUC of 0.6693 for POPF. DM was observed to be associated with a lower incidence of pancreatojejunostomy leakage, yet this association may be influenced by pancreatic texture and warrants cautious interpretation, pending validation in larger studies. Patients with POPF were more frequently observed to require reoperation during their postoperative course.

## Figures and Tables

**Figure 1 medicina-62-00094-f001:**
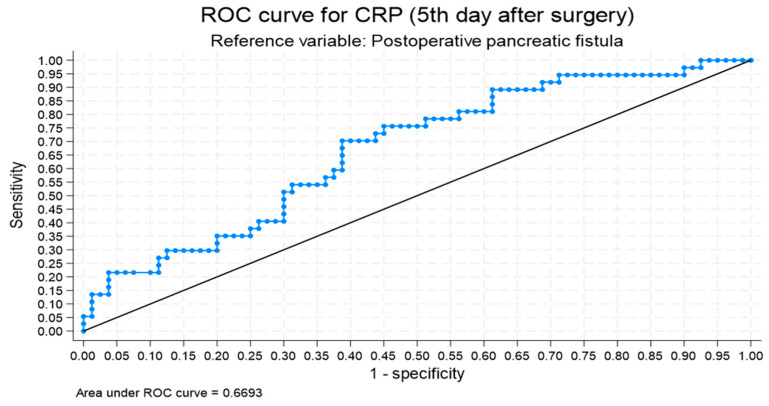
CRP on POD 5 exceeding 125.3 mg/dL is associated with the presence of postoperative pancreatic fistula (POPF). The CRP threshold has a sensitivity of 70.3% and a specificity of 61.3%, with an AUC of 0.669.

**Table 1 medicina-62-00094-t001:** Patient demographics regarding the preoperative numerical variables assessed.

Parameter	No POPF Mean (St Deviation)	POPF Mean (St Deviation)	Total Mean (St Deviation)	*p*
Age (years)	67.7 (9.2)	67.9 (9.0)	67.8 (9.1)	0.888
BMI (kg/m^2^)	26.6 (3.8)	26.4 (3.7)	26.6 (3.8)	0.784
Bilirubin (mg/dL)	6.0 (7.0)	6.8 (7.7)	6.3 (7.2)	0.601
Albumin (g/dL)	3.6 (0.6)	3.7 (0.6)	3.6 (0.6)	0.644
Ferritin (μg/L)	512.5 (534.9)	466.1 (615.4)	498.1 (558.8)	0.681
CA 19-9 (U/mL)	2520.2 (10,816.5)	2418.5 (6886.4)	2488.3 (9725.8)	0.958
HbA1c (%)	6.3 (1.2)	5.9 (0.8)	6.2 (1.1)	**0.037**
CEA (ng/mL)	5.2 (5.8)	4.4 (4.2)	4.9 (5.4)	0.494
PLT (10^3^/μL)	289.5 (103.3)	291.9 (73.4)	290.3 (94.7)	0.896
Duration of jaundice (days)	27.9 (41.4)	32.9 (45.9)	29.5 (42.7)	0.575
Smoking (ppys)	33.1 (37.4)	29.8 (33.4)	32.0 (36.0)	0.666

POPF: postoperative pancreatic fistula, BMI: body mass index, CA 19-9: carbohydrate antigen 19-9, CEA: carcinoembryonic antigen, PLT: platelets, ppy: pack per year. Bold values indicate statistically significant differences between groups (*p* < 0.05).

**Table 2 medicina-62-00094-t002:** Patient demographics regarding the categorical variables assessed.

Parameter	No POPF N (%)	POPF N (%)	Total N (%)	*p*
Sex (F/M)	25 (30.9) 56 (69.1)	11 (29.7) 26 (70.3)	36 (30.5) 82 (69.5)	0.901
Jaundice	64 (80.0)	31 (83.8)	95 (81.2)	0.626
Weight loss	72 (90.0)	27 (73.0)	99 (84.6)	**0.018**
Anorexia	26 (32.5)	11 (29.7)	37 (31.6)	0.764
Diabetes mellitus	32 (40.0)	6 (16.2)	38 (32.5)	**0.011**
Current smoking	37 (46.3)	15 (40.5)	52 (44.4)	0.563
Alcohol consumption	15 (18.8)	8 (21.6)	23 (19.7)	0.716
Cholecystectomy	14 (17.5)	4 (10.8)	18 (15.4)	0.351
Chronic pancreatitis	7 (8.6)	1 (2.7)	8 (6.8)	0.234
Neoadjuvant Chemotherapy	9 (11.1)	3 (8.1)	12 (10.2)	0.751
ERCP	51 (63.0)	26 (70.3)	77 (65.3)	0.439

POPF: postoperative pancreatic fistula, ERCP: endoscopic retrograde cholangiopancreatography. Bold values indicate statistically significant differences between groups (*p* < 0.05).

**Table 3 medicina-62-00094-t003:** Intraoperative categorical variables of patients undergoing pancreatoduodenectomy.

Parameter	No POPF N (%)	POPF N (%)	Total N (%)	*p*
**Pancreas texture**				**0.007**
Soft	29 (35.8)	22 (59.5)	51 (43.2)	
Hard	40 (49.4)	7 (18.9)	47 (39.8)	
Intermediate	12 (14.8)	8 (21.6)	20 (16.9)	
**Soft pancreas**	29 (35.8)	22 (59.5)	51 (43.2)	**0.016**
**Texture/Pancreatic duct Classification**				0.067
A: non-soft > 3 mm	35 (43.2)	10 (27.0)	45 (38.1)	
B: non-soft ≤ 3 mm	17 (21.0)	5 (13.5)	22 (18.6)	
C: soft > 3 mm	15 (18.5)	8 (21.6)	23 (19.5)	
D: soft ≤ 3 mm	14 (17.3)	14 (37.8)	28 (23.7)	
**Pancreas risk Score**				**0.002**
Negligible (0)	12 (14.8)	0 (0)	12 (10.1)	
Low risk (1–2)	34 (41.9)	10 (27)	44 (37.2)	
Intermediate Risk (3–6)	34 (41.9)	25 (67.5)	59 (50)	
High risk (≥7)	1 (1.2)	2 (5.4)	3 (2.5)	
**Blood transfusion**	47 (60.3)	21 (58.3)	68 (59.6)	0.846
**Pylorus preservation**	17 (21.0)	8 (21.6)	25 (21.2)	0.938
**Anastomotic technique**				0.065
(1) Duct to mucosa				
(2) Blumgart	31 (38.2)	18 (48.6)	49 (41.5)	
(3) Duct to mucosa with seromuscular	40 (49.3)	9 (24.3)	49 (41.5)	
jejunal flap formation	5 (6.1)	6 (16.2)	11 (9.3)	
(4) External wirsungostomy	5 (6.1)	3 (8.1)	8 (6.7)	
(5) Invagination PJ	0 (0.0)	1 (2.7)	1 (0.8)	
**Blumgart technique**	40 (49.4)	9 (24.3)	49 (41.5)	**0.010**

POPF: postoperative pancreatic fistula, PJ: pancreatojejunostomy. Bold values indicate statistically significant differences between groups (*p* < 0.05).

**Table 4 medicina-62-00094-t004:** Intraoperative numerical variables of patients undergoing pancreatoduodenectomy.

Parameter	No POPF Mean (St Deviation)	POPF Mean (St Deviation)	Total Mean (St Deviation)	*p*
Pancreatic duct diameter (mm)	4.0 (1.5)	3.4 (1.3)	3.8 (1.4)	0.053
Cutting surface horizontal diameter (cm)	2.6 (0.7)	2.8 (0.6)	2.7 (0.7)	0.116
Cutting surface vertical diameter (cm)	1.8 (0.7)	1.6 (0.5)	1.8 (0.6)	0.057
Horizontal/vertical ratio	1.8 (0.5)	2.0 (0.8)	1.9 (0.6)	0.138
Cutting surface area (cm^2^)	3.7 (1.8)	3.6 (1.5)	3.7 (1.7)	0.593
Blood units	1.0 (1.6)	0.9 (1.2)	1.0 (1.4)	0.778
Crystalloids (L)	6.5 (2.2)	6.3 (1.9)	6.4 (2.1)	0.734
Fresh frozen plasmas	1.4 (1.6)	1.2 (1.6)	1.3 (1.6)	0.505
Total number of lymph nodes (LNs)	23.7 (8.8)	22.9 (7.2)	23.4 (8.3)	0.628
Positive LNs	2.7 (3.5)	2.3 (2.7)	2.6 (3.2)	0.500
LN ratio	0.1 (0.1)	0.1 (0.1)	0.1 (0.1)	0.593
Duration of surgery (min)	371.2 (65.6)	366.2 (80.7)	369.7 (70.1)	0.736
Fistula risk score [14]	2.49 (1.747)	3.62 (1.754)	2.85 (1.82)	**0.002**

POPF: postoperative pancreatic fistula, LN: lymph nodes. Bold values indicate statistically significant differences between groups (*p* < 0.05).

**Table 5 medicina-62-00094-t005:** Postoperative course and outcomes of patients undergoing pancreatoduodenectomy.

Parameter	No POPF N (%)	POPF N (%)	Total N (%)	*p*
Reoperation	3 (3.8)	14 (37.8)	17 (14.5)	**<0.001**
Clavien–Dindo classification				
Minor (CD1-CD3A)	28 (84.8)	23 (62.2)	51 (72.9)	
Major (CD3B-CD5)	5 (15.2)	14 (37.8)	19 (27.1)	**0.033**
Use of somatostatin	61 (75.3)	27 (73.0)	88 (74.6)	0.787
30-day mortality	3 (3.7)	3 (8.1)	6 (5.1)	0.376
90-day mortality	7 (8.6)	8 (21.6)	15 (12.7)	**0.050**

POPF: postoperative pancreatic fistula, BL: biochemical leak. Bold values indicate statistically significant differences between groups (*p* < 0.05).

**Table 6 medicina-62-00094-t006:** Histology and TNM classification of patients undergoing pancreatoduodenectomy.

Parameter	No POPF N (%)	POPF N (%)	Total N (%)	*p*
**Disease**				0.127
PDAC	62 (76.5)	21 (56.7)	83 (70.3)	
Ampulla of Vater AC	8 (9.8)	7 (18.9)	15 (12.7)	
Bile duct AC	5 (6.1)	3 (8.1)	8 (6.8)	
Pancreatic NET	1 (1.2)	2 (5.4)	3 (2.5)	
Duodenal AC	1 (1.2)	2 (5.4)	3 (2.5)	
Chronic pancreatitis	2 (2.4)	0 (0.0)	2 (1.7)	
Pancreatic cyctadenoma	0 (0.0)	1 (2.7)	1 (0.8)	
IPMN	1 (1.2)	0 (0.0)	1 (0.8)	
Ampulla of Vater NEC	1 (1.2)	0 (0.0)	1 (0.8)	
Gastric AC	0 (0.0)	1 (2.7)	1 (0.8)	
PDAC	62 (76.5)	21 (56.7)	83 (70.3)	**0.049**
Positive lymph nodes	53 (65.4)	20 (54.1)	73 (61.9)	0.238
**TNM Staging**				0.429
IA	1 (1.3)	3 (8.1)	4 (3.4)	
IB	17 (21.3)	10 (27.0)	27 (23.1)	
IIA	7 (8.8)	4 (10.8)	11 (9.4)	
IIB	28 (35.0)	9 (24.3)	37 (31.6)	
III	21 (26.3)	7 (18.9)	28 (23.9)	
IV	2 (2.5)	2 (5.4)	4 (3.4)	
No Cancer	4 (5.0)	2 (5.4)	6 (5.1)	
**Stage < IIB**				
No	50 (61.7)	19 (51.4)	69 (58.5)	
Yes	27 (33.3)	16 (43.2)	43 (36.4)	0.598
No cancer	4 (4.9)	2 (5.4)	6 (5.1)	0.283

PDAC: pancreatic ductal adenocarcinoma, AC: adenocarcinoma, NET: neuroendocrine tumor, IPMN: intraductal papillary mucinous neoplasm, NEC: neuroendocrine carcinoma. Bold values indicate statistically significant differences between groups (*p* < 0.05).

**Table 7 medicina-62-00094-t007:** Inflammatory markers and amylase of plasma and drain in the postoperative period.

	No POPF Mean (St Deviation)	POPF Mean (St Deviation)	Total Mean (St Deviation)	*p*
WBC count POD 1 (×10^3^/uL)	11.8 (4.0)	11.9 (4.4)	11.8 (4.1)	0.865
WBC count POD 3 (×10^3^/uL)	10.5 (3.8)	11.2 (3.8)	10.7 (3.8)	0.404
WBC count POD 5 (×10^3^/uL)	9.3 (3.3)	10.0 (3.2)	9.5 (3.3)	0.312
CRP POD 1 (mg/L)	109.9 (46.6)	131.3 (39.4)	116.5 (45.4)	**0.018**
CRP POD 3 (mg/L)	185.7 (82.4)	221.8 (58.8)	197.4 (77.2)	**0.009**
CRP POD 5 (mg/L)	121.8 (78.9)	171.9 (88.6)	137.6 (84.9)	**0.003**
PLT POD 1 (×10^3^/uL)	248.7 (91.7)	241.9 (65.4)	246.6 (84.2)	0.689
PLT POD 3 (×10^3^/uL)	235.7 (102.0)	227.4 (79.9)	233.1 (95.3)	0.666
PLT POD 5 (×10^3^/uL)	291.2 (138.5)	286.8 (127.5)	289.7 (134.3)	0.871
Serum amylase POD 0 (U/L)	160.6 (187.9)	269.0 (184.5)	193.6 (192.7)	**0.005**
Serum amylase POD 1 (U/L)	230.3 (356.0)	307.8 (207.4)	254.9 (317.4)	0.235
Serum amylase POD 2 (U/L)	105.3 (155.7)	133.6 (102.1)	114.9 (140.0)	0.320
Drain amylase POD 1 (U/L)	591.0 R (1383.6) 4438.2 L (13,277.8)	2025.5 R (3544.4) 6978.7 L (10,649.1)	1018.8 R (2332.7) 5225.1 L (12,531.0)	**0.028** 0.321
Drain amylase POD 3 (U/L)	323.6 R (1042.1) 1068.5 L (3288.1)	1441.9 R (4302.5) 2207.3 L (3117.0)	679.4 R (2604.9) 1451.6 L (3262.3)	0.138 0.084
Drain amylase POD 5 (U/L)	150.5 R (732.3) 774.3 L (2088.6)	442.6 R 3950.8 L	246.1 R 1775.9 L	0.221 **0.016**

POPF: postoperative pancreatic fistula, WBC: white blood cell, CRP: C-reactive protein, PLT: platelet, POD: postoperative day, L: left, R: right. Bold values indicate statistically significant differences between groups (*p* < 0.05).

**Table 8 medicina-62-00094-t008:** Univariate logistic regression analysis. Dependent variable: Postoperative pancreatic fistula.

Parameter	Odds Ratio	95% Confidence Interval	*p*
Univariate Analysis			
**DM**	0.290	0.108–0.775	**0.014**
**Serum amylase POD 0 (U/L)**	1.003	1.000–1.005	**0.008**
**CRP POD 5 (mg/L)**	1.007	1.002–1.011	**0.004**
Age (years)	1.003	0.961–1.047	0.887
Sex	1.055	0.452–2.464	0.901
Histology (PDAC vs. other pathology)	0.449	0.195–1.034	0.060
Pancreatic cutting surface Area (cm^2^)	0.936	0.737–1.190	0.591
Horizontal to vertical pancreatic cutting surface ratio	1.599	0.831–3.076	0.160
Pancreatic duct diameter (mm)	0.747	0.553–1.008	0.056
**Pancreas texture (hard/intermediate vs. soft)**	0.380	0.171–0.845	**0.018**
Lymph nodes	0.621	0.281–1.372	0.239
Ca19-9 (U/mL)	1.000	0.999–1.000	0.985
Duration of surgery (minutes)	0.999	0.993–1.004	0.733
History of smoking	0.792	0.389–1.746	0.564
Weight loss	0.925	0.851–1.004	0.063

Multiple model calibration: Hosmer–Lemeshow χ^2^ = 5.55, *p* = 0.136; multiple model discrimination: area under ROC curve (AUC) = 0.723. CRP: C-reactive protein, POD: postoperative day, POPF: postoperative pancreatic fistula, BMI: body mass index, CRP: C-reactive protein, DM: diabetes mellitus, PDAC: pancreatic ductal adenocarcinoma. Bold values indicate statistically significant differences between groups (*p* < 0.05).

**Table 9 medicina-62-00094-t009:** Multivariate logistic regression analysis.

Multivariate Analysis			
Parameter	Odds Ratio	95% Confidence Interval	*p*
CRP POD 5	1.007	1.001–1.012	**0.025**
DM	0.254	0.084–0.763	**0.015**
Pancreas texture (hard/intermediate vs. soft)	1.747	0.708–4.311	0.226
Serum amylase POD 0	1.002	0.999–1.004	0.149

Dependent variable: Postoperative pancreatic fistula. CRP: C-reactive protein, POD: postoperative day, DM: diabetes mellitus. Bold values indicate statistically significant differences between groups (*p* < 0.05).

## Data Availability

Dataset available on request from the authors.

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
