# Peer review of "Medicina2026, 62(1), 94;https://doi.org/10.3390/medicina62010094"

_medicina, 2026, doi:10.3390/medicina62010094_

Round 1

Reviewer 1 Report

Comments and Suggestions for Authors

Contemporary assessment of post-operative pancreatic fistula after pancreatoduodenectomy in a European Hepato-Pancreato-Biliary Center: a 5-year experience (medicina-3941575)

The authors have put together a comprehensive assessment of POPF in 118 patients over a period of ~5 years that provides a good foundation for statistical analysis. The study set up is of clinical relevance and focuses on practical, readily measured predictors for postoperative monitoring. The authors have properly discussed their results in the context of existing literature in this field. It provides a valuable contribution to the ongoing understanding of POPF risk stratification and postoperative management following PD. However, the study has few weaknesses, which the authors must address for a better understanding.

Major:

  1. The authors should discuss the applicability of their study parameters on a global scale. Would their study outcomes be discernible to other institutions using different surgical techniques, patient demographics, or perioperative management protocols?
  2. Since only 37 out of 118 patients developed POPF, it reduces the statistical power of the multivariate model as the sample size for subgroup analysis is relatively small and may limit the robustness of the conclusions. How do authors explain this point?
  3. The authors have listed the AUC values of 0.669 for CRP and 0.717 for amylase which seem to be of poor diagnostic accuracy and could make these markers not reliable as standalone predictive tools. The authors should discuss this point.

Minor: The manuscript is well-formatted for most part, a few grammatical inconsistencies and minor formatting issues are present such as use of full form of Pancreatoduodenectomy (PD) in section 2.4, for example, despite abbreviating in the first appearance in Introduction, line 1. The authors can make such abbreviations consistent across the manuscript.

Decision: Minor revision is required. The authors should add discussion to the concerned sections pertaining to the above listed comments, which would enhance the manuscript's overall impact on the audience.

Author Response

Thank you very much for your valuable comments on our manuscript. We thoroughly reviewed your comments and made the adjustments/clarifications as per your request.

C: The authors should discuss the applicability of their study parameters on a global scale. Would their study outcomes be discernible to other institutions using different surgical techniques, patient demographics, or perioperative management protocols

A: We thank the reviewer for this insightful comment. We agree that the external applicability of our findings is an important consideration. Our study was conducted in a medium-volume European HPB center, where surgical and perioperative management protocols are standardized. Although these protocols may vary internationally, the identified predictors of POPF—namely postoperative CRP, serum amylase, and the presence of diabetes mellitus—are biochemical and clinical parameters that are universally measurable and not dependent on specific surgical techniques.

Nonetheless, we acknowledge that differences in patient demographics, pancreatic pathology prevalence, and perioperative management across institutions may influence absolute POPF rates. To address this, we have added a statement in the Discussion section (12th paragraph)  emphasizing that our findings should be validated in multicentric or international cohorts to confirm their generalizability (Although, our study was conducted in a single....)

C: Since only 37 out of 118 patients developed POPF, it reduces the statistical power of the multivariate model as the sample size for subgroup analysis is relatively small and may limit the robustness of the conclusions. How do authors explain this point?

A: Thank you for this valuable observation. We acknowledge that the number of POPF cases (n = 37) may limit the statistical power of multivariate modeling. To minimize the risk of overfitting, we included only variables that were both statistically significant in univariate analysis and clinically relevant (CRP on POD 5, diabetes mellitus, serum amylase POD0 and soft pancreas). This resulted in an events-per-variable (EPV) ratio of approximately 18, which exceeds the conventional threshold of ≥10 events per variable recommended for stable logistic regression models [31,32]. Therefore, we believe that the model maintains acceptable internal validity despite the moderate sample size. Nevertheless, we have added a statement in the Discussion acknowledging that the relatively small number of POPF events may limit the external generalizability of our findings and that further multicenter studies are warranted. Reoperation as a predictor variable was omitted, according to another reviewer’s suggestion.

References in the manuscript

31. Peduzzi P, Concato J, Kemper E, Holford TR, Feinstein AR. A simulation study of the number of events per variable in logistic regression analysis. J Clin Epidemiol. 1996;49(12):1373-1379.

32. Vittinghoff E, McCulloch CE. Relaxing the rule of ten events per variable in logistic and Cox regression. Am J Epidemiol. 2007;165(6):710-718.

The modifications can be found in the 16th paragraph of the Discussion section ( Although only 37 patients developed POPF...).

C: The authors have listed the AUC values of 0.669 for CRP and 0.717 for amylase which seem to be of poor diagnostic accuracy and could make these markers not reliable as standalone predictive tools. The authors should discuss this point.

A:  We appreciate the reviewer’s important point. We agree that the AUC values of 0.669 for CRP and 0.717 for amylase indicate moderate, rather than high, diagnostic accuracy. These markers should not be interpreted as standalone diagnostic tools, but rather as adjunctive parameters to aid early clinical suspicion of POPF in conjunction with other findings such as intraoperative pancreas measurements, clinical presentation and imaging studies We have now clarified in the Discussion that these biomarkers serve as useful screening indicators, helping to guide further diagnostic evaluation rather than providing definitive diagnosis. It is mentioned in the middle of the 9th paragraph of the Discussion section (While CRP and serum amylase demonstrated...) 

C: The manuscript is well-formatted for most part, a few grammatical inconsistencies and minor formatting issues are present such as use of full form of Pancreatoduodenectomy (PD) in section 2.4, for example, despite abbreviating in the first appearance in Introduction, line 1. The authors can make such abbreviations consistent across the manuscript.

A: Thanks for the comment. We thoroughly scanned the manuscript and made the corrections regarding the abbreviations.

We hope that we managed to answer and adjust all of your valuable comments regarding our submitted manuscript

With regards

The authors of the manuscript  

Reviewer 2 Report

Comments and Suggestions for Authors

This is a retrospective single-centre study with 118 consecutive patients undergoing pancreatoduodenectomy (PD) between 2018–2024. The aim of the study is to identify predictors of POPF. The authors proposed inclusion of biochemical markers such as CRP and amylase as novel predictors of POPF. I found no novelty in this study and certain methodological aspects are concerning. Here is a list of some of my major concerns:

  1. The findings in this study are well known and there is stronger evidence in literature that already established associations between risk factors such as soft gland, small duct, CRP, diabetes with POPF long time ago. There is a number of recent large multicentre analyses or meta-analyses in the literature  e.g. ANZ J Surg 2021. It would have been more useful to learn what the authors would do at their centre to mitigate the predicted risks of High risk POPF. 
  2. The authors have referenced the 2016 ISGPS definition of POPF.  The authors must confirm that all cases were re-classified according to the updated criteria, especially distinguishing biochemical leak from Grade B/C POPF.
  3. The logistic regression model includes numerous variables for only 37 POPF events. I find this very concerning about model overfitting. The authors should state how variables were selected (stepwise, backward elimination, etc.) and report model calibration metrics. The presentation of Table 8 mixes univariate and multivariate results inconsistently. I find it very irritating that the confidence intervals and decimal formatting were inconsistent (commas vs points). These should be standardised.
  4. The conclusion that diabetes reduces POPF risk is somewhat misleading and likely confounded by factors such as small cohort study population and pancreatic texture. Such strong conclusion cannot be reached without strong and robust data analysis which is a definite weakness in this small study. There is no mechanistic explanation offered by the authors at all to support such conclusion.
  5. Figure 2 is very simplistic and adds no clinical value. If anything, I would suggest that the authors depict how CRP integrates with drain amylase and clinical signs to guide imaging or intervention.

Author Response

Thank you very much for your valuable comments on our manuscript. We thoroughly reviewed your comments and made the adjustments/clarifications as per your request.

C:The findings in this study are well known and there is stronger evidence in literature that already established associations between risk factors such as soft gland, small duct, CRP, diabetes with POPF long time ago. There is a number of recent large multicentre analyses or meta-analyses in the literature  e.g. ANZ J Surg 2021. It would have been more useful to learn what the authors would do at their centre to mitigate the predicted risks of High risk POPF. 

A: We thank the reviewer for this constructive comment and fully acknowledge that many of the risk factors identified in our study have been previously established by larger multicentric studies and meta-analyses (including Kamarajah et al., ANZ J Surg 2021; Russell & Aroori, 2022). The purpose of our analysis was not to redefine known risk factors but to validate their applicability in a contemporary European single-centre context, characterized by consistent surgical techniques and postoperative care protocols.

We agree that the practical implications of our findings should be emphasized. In our centre, patients identified as high-risk for POPF (soft pancreas, small pancreatic duct, CRP >125 mg/L on POD 5, or absence of diabetes) are now managed with a structured mitigation strategy including: Early imaging (CT scan) when CRP exceeds 125 mg/L, prolonged drainage and selective use of somatostatin analogs, delayed initiation of oral intake and intensified monitoring in high-dependency units and early multidisciplinary review (surgeon, interventional radiologist, infectious disease specialist and dietitians).

We have expanded the Discussion to describe this institutional approach and its clinical rationale (11th paragraph in the discussion section - While our findings corroborate well-established....)

C: The authors have referenced the 2016 ISGPS definition of POPF.  The authors must confirm that all cases were re-classified according to the updated criteria, especially distinguishing biochemical leak from Grade B/C POPF

A:  We appreciate this important clarification. We confirm that all cases in our series were retrospectively re-evaluated and reclassified according to the 2016 updated ISGPS definition. In our dataset, biochemical leaks (BL) were explicitly separated from clinically relevant fistulas (Grade B and C), and only Grade B and C POPF were included in the main statistical analyses. POPF in the results and tables refers to Grade B and C. We have clarified this point in the Methods section to ensure transparency. (Material and Methods- Definitions used in the study- middle of the paragraph "All postoperative pancreatic fistulas were retrospectively reclassified according to the 2016 ISGPS update...)

C: I find it very irritating that the confidence intervals and decimal formatting were inconsistent (commas vs points). These should be standardised.

A: We apologize for that. In the revized version, commas have been replaced with points in all tables.

C: The conclusion that diabetes reduces POPF risk is somewhat misleading and likely confounded by factors such as small cohort study population and pancreatic texture. Such strong conclusion cannot be reached without strong and robust data analysis which is a definite weakness in this small study. There is no mechanistic explanation offered by the authors at all to support such conclusion

A: We thank the reviewer for highlighting this point and agree that the observed inverse association between diabetes and POPF must be interpreted with caution. We have revised both the Discussion and Conclusion to present this as an observed association rather than a definitive causal relationship. We now clarify that the protective association may be confounded by pancreatic texture, as diabetic patients more frequently present with chronic fibrosis and reduced exocrine function—both of which lower the likelihood of clinically significant leakage. This biological rationale has been previously suggested by Mathur et al. (Ann Surg 2007) and Xia et al. (HPB Dis Int 2015), and we have incorporated these references to support a mechanistic explanation.  Discussion- 5th paragraph "The apparent protective effect of diabetes on POPF observed in our cohort..." and in the Conclusions "yet this finding may be influenced by pancreatic texture and warrants cautious interpretation pending validation in larger studies".

We also changed the Conclusion of the abstract to: "POPF remains an important complication after PD. CRP >126 mg/dL on POD 5 is an important predictor for POPF and this finding should trigger review with cross-sectional imaging. Diabetes mellitus was associated with a lower incidence of pancreatojejunostomy leakage in this cohort; however, this finding should be interpreted cautiously given the study’s sample size and potential confounding factors."

C: Figure 2 is very simplistic and adds no clinical value. If anything, I would suggest that the authors depict how CRP integrates with drain amylase and clinical signs to guide imaging or intervention.

A: We thank the reviewer for this valuable observation. We agree that Figure 2, in its previous form, provided limited clinical value and did not effectively illustrate the integration of CRP, drain amylase, and clinical parameters in guiding management decisions. Accordingly, we have decided to remove this figure from the revised version of the manuscript.

C: The logistic regression model includes numerous variables for only 37 POPF events. I find this very concerning about model overfitting. The authors should state how variables were selected (stepwise, backward elimination, etc.) and report model calibration metrics. The presentation of Table 8 mixes univariate and multivariate results inconsistently.

A: We thank the reviewer for these detailed and constructive remarks.
1) To minimize overfitting, we applied a backward stepwise logistic regression method with entry criterion p < 0.05 and removal criterion p > 0.10. Only variables that were both statistically significant in univariate analysis and clinically relevant were considered for entry. The final model retained 4 independent predictors (CRP on POD 5, diabetes mellitus,serum amylase on POD0 and soft pancreas), achieving an events-per-variable ratio of 18, which meets the accepted threshold for logistic modeling stability [31,32] (references can be found at the end of the revised manuscript).

2) Model calibration was assessed using the Hosmer–Lemeshow goodness-of-fit test (χ² = 5.55, p = 0.136), indicating good calibration. The model’s discrimination was measured by the area under the ROC curve (AUC = 0.723), reflecting satisfactory predictive performance. These details have been added to the revised Methods and Results sections.
3) Table 8 has been reformatted to clearly separate univariate and multivariate analyses (Now Table 8 and Table 9).

Reoperation as a predictor variable was omitted, according to another reviewer’s suggestion, suggesting to run the uni and multivariate analysis again.

In the manuscript, modifications can be found in the 3rd paragraph of the Statistical Analysis section (Univariate logistic regression was initially performed to evaluate...).

Univariate and multivariate analysis have been separated as two independent tables (8 and 9). The univariate and multivariate analysis paragraphs have been adjusted accordingly. (i.e in the multivariate analysis paragraph " The final multivariate model retained four variables: CRP on POD 5, diabetes mellitus...)

We hope that we managed to answer and adjust all of your valuable comments regarding our submitted manuscript

With regards

The authors of the manuscript  

Reviewer 3 Report

Comments and Suggestions for Authors

The present study assesses the incidence of POPF after PD in a single-center institutional experience. Furthermore, potential predictors of POPF occurrence are explored. Univariate analyses identified low HbA1c, weight loss, absence of diabetes, soft pancreas, small Wirsung duct, other than the Blumgart technique, other than PDAC pathology, postoperative CRP, and serum amylase and drain amylase as predictors of POPF occurrence after PD. Interestingly, the ISGPS risk classification for POPF after PD did not emerge as a significant predictor, as one might expect. Although there is no novelty in the field with this study (there are many other studies addressing the same topic with much more patients included in the analyses than in the present cohort), the results could add value to the existing literature on POPF after PD.

A few concerns should be raised before considering the acceptance:

The number of analyzed patients is relatively low, considering that the center is an HPB one. Thus, with less than 25 PD/year, the center should be classified as a low to medium-volume center. This aspect should be stated as a limitation of the study and may help explain, at least in part, the relatively high POPF and mortality rates. It would be interesting to provide the number of surgeons who performed PDs during the analyzed period to examine whether there is an association between the severity of POPF, mortality rates, and surgeon case load.

In Table 1, there is a parameter called 'survival'. What is the relevance of this parameter as a potential predictor of POPF occurrence, as this is a postoperative long-term outcome? Please consider eliminating this parameter from the analyses. Furthermore, the study included PDs across all indications, which have different long-term survival rates. Nevertheless, no follow-up endpoint is provided. The same issue for postoperative discharge, which is obviously longer for patients with POPF (and not a potential predictor but a consequence).

How do the authors explain the uncommonly high CA 19-9 serum levels before surgery, which are at odds with the majority of studies published so far? In this context, why did such a small number of patients undergo neoadjuvant therapy, knowing that a CA 19-9 serum level of more than 500 is rather an indication for neoadjuvant therapy?

In Tables 1 and 2, weight loss is included twice. Furthermore, in Table 1, there is no statistical significance between the POPF and no-POPF groups, whereas in Table 2 there is. Please be consistent and explain the differences.

Why did the authors include only PLT and not also HB and leucocytes in the comparative analyses?

Please consider including in the analyses other important factors that could influence the occurrence of POPF, such as preoperative biliary drainage or cholangitis.

The POPF risk score (as proposed by Callery MP et al.) should also be included in the analyses to stratify patients at risk of developing POPF.

The inclusion of the POPF classification in Table 5 is meaningless, along with the p-value.

In Table 6, it is stated that it is unclear what the percentages of patients with and without POPF are in the group of patients with PDAC. (76.5% vs. 56.7% is more than 100%!). Please clarify.

Potential inflammatory markers should also be assessed preoperatively, as a few patients might have elevated inflammatory markers in the preoperative setting due to preoperative biliary drainage or cholangitis.

The increased reoperation rate in patients with POPF is a consequence of POPF, not a predictor. Thus, it should be excluded from both univariate and multivariate analyses for potential predictors of POPF occurrence after PD. In the multivariate analyses, only factors identified as statistically significant in the univariate analyses should be included. Thus, Table 8 should be entirely redone.

Minor issues:

Please be consistent in using the abbreviations throughout the manuscript.

Author Response

Thank you very much for your valuable comments on our manuscript. We thoroughly reviewed your comments and made the adjustments/clarifications as per your request.

C: The number of analyzed patients is relatively low, considering that the center is an HPB one. Thus, with less than 25 PD/year, the center should be classified as a low to medium-volume center. This aspect should be stated as a limitation of the study and may help explain, at least in part, the relatively high POPF and mortality rates. It would be interesting to provide the number of surgeons who performed PDs during the analyzed period to examine whether there is an association between the severity of POPF, mortality rates, and surgeon case load.

A: We thank the reviewer for this important observation. We agree that our center’s annual PD volume classifies it as a medium-volume HPB unit, and this factor could indeed influence both POPF incidence and perioperative outcomes. This is now clearly acknowledged in the Limitations section. Regarding the surgical team, PDs during the study period were performed by four consultant surgeons, each contributing to a portion of the total caseload. We performed an exploratory comparison between individual surgeon case volume and POPF occurrence but found no statistically significant association; however, due to the limited number of cases per surgeon, these results were not powered for formal analysis. 

Discussion- 15th paragraph "Our center performs approximately 25 PD annually, classifying..."

C: In Table 1, there is a parameter called 'survival'. What is the relevance of this parameter as a potential predictor of POPF occurrence, as this is a postoperative long-term outcome? Please consider eliminating this parameter from the analyses. Furthermore, the study included PDs across all indications, which have different long-term survival rates. Nevertheless, no follow-up endpoint is provided. The same issue for postoperative discharge, which is obviously longer for patients with POPF (and not a potential predictor but a consequence).

A: Thanks for your comment. Indeed, survival is not supposed to be a predictor for POPF. Moreover, the length of stay is longer in patients with CR POPF. it is not supposed to be a predictor rather a consequence of POPF as those patients experienced more complications than the control group. We removed the parameters Survival and Length of Stay.

We also clarified that these parameters were analyzed descriptively as outcomes, not as predictors. (1st paragraph of Statistical Analysis section) Moreover, we revised the Results section to emphasize that the association between POPF and longer hospital stay represents a postoperative consequence rather than a risk factor. (postoperative outcome section: "As expected, patients who developed POPF experienced...)

Finally, we added a clarifying statement that follow-up duration and survival were not prespecified study endpoints, as the analysis was focused solely on perioperative predictors of POPF. (Final paragraph of the discussion section)

C: How do the authors explain the uncommonly high CA 19-9 serum levels before surgery, which are at odds with the majority of studies published so far? In this context, why did such a small number of patients undergo neoadjuvant therapy, knowing that a CA 19-9 serum level of more than 500 is rather an indication for neoadjuvant therapy?

A:  We thank the reviewer for highlighting this point. We acknowledge that the mean CA 19-9 levels in our cohort were higher than those reported in most large studies. This likely reflects the high proportion of patients with obstructive jaundice and delayed preoperative biliary drainage, both of which can artificially elevate CA 19-9 values. 

The decision of proceeding to upfront surgery vs neoadjuvant chemotherapy was mostly based on anatomic criteria of the tumor on ct scan images (whether the tumor was resectable or not) , and the performance status of the patients. Indeed, in the ABC criteria for borderline resectable PDAC, ca 19-9 plays an important role, yet in the MDT meetings the first two factors were mostly assessed.

C: In Tables 1 and 2, weight loss is included twice. Furthermore, in Table 1, there is no statistical significance between the POPF and no-POPF groups, whereas in Table 2 there is. Please be consistent and explain the differences.

A: We thank the reviewer for this accurate observation. The apparent duplication results from the variable weight loss being analyzed in two different ways: As a continuous variable (expressed in kilograms lost preoperatively) in Table 1.As a binary categorical variable in Table 2, following the clinical definitions used in our database. The numerical analysis (Table 1) did not reach statistical significance, whereas the categorical classification showed significance because the distribution of patients meeting the weight loss criterion differed between groups.

To avoid confusion, we have now retained only one consistent form of the variable across all tables—presented as a binary categorical variable (“presence of clinically significant weight loss: yes/no”)—and have updated Tables 1 and 2 accordingly. The text of the Results section has also been revised for consistency.

Α clarification was made in the discussion section (Discussion, 7th paragraph, "Preoperative weight loss was observed..." Also a definition of unintentional weight loss was given in the last sentence on the 'Definitions used in the study section".

C: Why did the authors include only PLT and not also HB and leucocytes in the comparative analyses? Please consider including in the analyses other important factors that could influence the occurrence of POPF, such as preoperative biliary drainage or cholangitis.

A:  We included in the analysis several factors that may serve as inflammatory markers, aiming to identify patients with an ongoing or potential inflammatory process and possible sepsis related to POPF. For this reason, we analyzed CRP and platelet (PLT) count, both of which have been investigated in the literature as potential predictive factors for POPF. Leukocyte (white blood cell, WBC) count was also included in our analysis, as shown in Table 7. However, WBC count was not found to be a statistically significant factor associated with POPF.

In table 2 it is mentioned that 77 patients had preoperative ERCP (65.3%). Specifically, 51 out of 81 without POPF (63%) and 26 out of 37(70.3%) had a preoperative biliary drainage with brushings. It was not statistically significant in our analysis (p=0.439). The reason for ercp was mostly for diagnosis (imaging and cytology), decompression of the biliary tree prior to neoadjuvant chemotherapy, normalization of the bilirubin prior to PD as septic complications, SSI, and liver failure have been reported as possible postoperative complications in jaundiced patients. Cholangitis as an entity was present in quite a few patients hence it was not included in the analysis.

C: The POPF risk score (as proposed by Callery MP et al.) should also be included in the analyses to stratify patients at risk of developing POPF.

A: We thank the reviewer for this suggestion. The Fistula Risk Score (FRS) by Callery et al. was retrospectively calculated for all patients using available intraoperative data. The mean FRS was 2.85 ± 1.82, consistent with intermediate risk. We have now incorporated this variable into Table 4 which showed a significant association with POPF occurrence (p = 0.002). We also included in table 3 the pancreas risk score from the same paper. This addition strengthens our findings and aligns them with established risk stratification systems.

C: The inclusion of the POPF classification in Table 5 is meaningless, along with the p-value.

A: It was removed from the table

C: In Table 6, it is stated that it is unclear what the percentages of patients with and without POPF are in the group of patients with PDAC. (76.5% vs. 56.7% is more than 100%!). Please clarify.

A: PDAC was present in our study in 62 out of 81 (62/81=0.765, 76.5%) patients without POPF and in 21 out of 37  21/37=0.567, 56.7%) patients with POPF. In total it was present in 83 out of 118 patients (70.3%).

C: Potential inflammatory markers should also be assessed preoperatively, as a few patients might have elevated inflammatory markers in the preoperative setting due to preoperative biliary drainage or cholangitis.

A: Thanks for the comment. It is true that preoperative inflammatory markers may be useful to assess the possible inflammatory process prior to resection, as some of the postoperative results may be affected. We did not routinely measured for example procalcitonin or crp, hence, as many data may be missing, we did not include them in the analysis. We wanted in our analysis to include variables with no missing data, given the fact that our sample (118 patients) is not very large and the number of parameters already addressed are quite a lot.

C: Please be consistent in using the abbreviations throughout the manuscript.

A Thanks for the comment. All abbreviations were corrected accordingly.

C:  The increased reoperation rate in patients with POPF is a consequence of POPF, not a predictor. Thus, it should be excluded from both univariate and multivariate analyses for potential predictors of POPF occurrence after PD. In the multivariate analyses, only factors identified as statistically significant in the univariate analyses should be included. Thus, Table 8 should be entirely redone.

A: We thank the reviewer for this critical observation and fully agree. In the initial version, reoperation was inadvertently included in the logistic regression model, although it represents a postoperative outcome rather than a potential predictor of POPF. We have now re-performed the regression analysis, excluding reoperation from both the univariate and multivariate models. Only variables that were statistically significant in univariate analysis (p < 0.10) were included in the revised multivariate model.

The new analysis identified the following independent predictors of POPF:

  • CRP on postoperative day 5 and Diabetes
  • The revised Table 8 and 9  now reflect only these legitimate predictors.

We have also clarified in the Methods section that multivariate modeling was performed using variables significant at the univariate level (p < 0.10) and restricted to pre- and intra-operative parameters to preserve temporal validity and avoid overfitting.

We hope that we managed to answer and adjust all of your valuable comments regarding our submitted manuscript

With regards

The authors of the manuscript  

Reviewer 4 Report

Comments and Suggestions for Authors

I am very happy to have had the opportunity to evaluate this excellent work and contribute to it. Kudos to the authors; it truly is a high-level article, with its figures and proposed management algorithm. I am particularly considering printing out Figure 2 and placing it on my board for use in our daily practice. This publication on postoperative pancreatic fistulas is educational for everyone, from experts to residents, and I really appreciated it. Additionally, I really appreciated the discussion section you prepared on diabetes and the likely explanation of its protective effect. I would just like to offer, humbly, a few suggestions to the authors to further improve the article:

1- On what basis did you assess the pancreas as soft, intermediate, or hard? I assume it was likely based on the surgeon’s intraoperative examination findings, but I would appreciate it if you could share a more objective assessment, if available. Including this explanation in the Methods section of the paper would make it look more polished, as this constitutes one of the meaningful outcomes.

2- Regarding PJ anastomosis techniques, especially the Blumgart technique, as expected from both the literature and our daily surgical practice, it yields good results in terms of fistula formation. The p-value was significant, yet the surgical technique was not discussed in the Discussion section. Including this could provide valuable insight for the readers, particularly for practicing surgeons.

3- Although it did not reach significance in the multivariate analysis, I think the cut-off value you identified for postoperative day 0 amylase could be important when monitoring patients on that day. Additionally, even though you recommended assessing amylase in the algorithm in Figure 2, this was not mentioned in the Discussion section. If you could share the authors’ perspective on this in a section of the Discussion, it would make the article more explanatory and informative.

I will be eagerly looking forward to evaluating your study after the revisions. With my deepest respect.    

Author Response

Thank you very much for your valuable comments on our manuscript. We thoroughly reviewed your comments and made the adjustments/clarifications as per your request.

C: On what basis did you assess the pancreas as soft, intermediate, or hard? I assume it was likely based on the surgeon’s intraoperative examination findings, but I would appreciate it if you could share a more objective assessment, if available. Including this explanation in the Methods section of the paper would make it look more polished, as this constitutes one of the meaningful outcomes.

A: The texture of the pancreas was assessed intraoperatively by the surgical team prior to performing the pancreatojejunal anastomosis. We acknowledge that intraoperative assessment can occasionally be subjective. Moreover, several preoperative imaging methods have been reported in the literature for evaluating the pancreas before surgery (Kalayarasan R. et al., 2023), and other indices have been proposed, such as the Pancreatic Attenuation Index (PAI) by Yardimci S. et al. (2015). Although factors such as peripancreatic fat tissue and fibrosis were described by our radiologists in some CT scans of patients undergoing pancreatoduodenectomy, this was not done systematically; therefore, it could not be included as part of our methods for assessing pancreatic texture.

This description has been added to the Materials and Methods section, within the Data Collection paragraph (Specifically, the texture of the pancreas…”).

C:  Regarding PJ anastomosis techniques, especially the Blumgart technique, as expected from both the literature and our daily surgical practice, it yields good results in terms of fistula formation. The p-value was significant, yet the surgical technique was not discussed in the Discussion section. Including this could provide valuable insight for the readers, particularly for practicing surgeons.

A: Regarding the anastomotic technique, it has been reported that the Blumgart anastomosis may be applicable to all patients in whom the pancreatic duct can be identified. This technique is associated with very low rates of significant postoperative morbidity and mortality, supporting its routine use for pancreatojejunal reconstruction after PD (Grobmyer S.R. et al., 2010). In our series, a total of five different types of anastomoses were used, depending on the surgeon’s preference and intraoperative findings (pancreatic texture and duct diameter). In the Discussion section, we added a paragraph (2nd paragraph) addressing the anastomotic techniques used, based on our findings and supporting evidence from the literature (“Multiple techniques in anastomosing the pancreas…”).

C: Although it did not reach significance in the multivariate analysis, I think the cut-off value you identified for postoperative day 0 amylase could be important when monitoring patients on that day. Additionally, even though you recommended assessing amylase in the algorithm in Figure 2, this was not mentioned in the Discussion section. If you could share the authors’ perspective on this in a section of the Discussion, it would make the article more explanatory and informative.

A: Thank you for the comment regarding the possible use of serum amylase on Day 0 for closer patient monitoring. The role of serum amylase as a predictor of POPF has been reported in the literature (van Dongen J.C. et al., 2021). In our multivariate analysis, serum amylase was not found to be statistically significant; therefore, it could not be included as a possible factor on our initially proposed algorithm. The amylase values on Day 3 refer to drain amylase levels, not serum levels, measured on postoperative Day 3, which is standard practice in our institution. This measurement is required to establish the diagnosis of pancreatic fistula according to the ISGPS definition (2016). However, this does not necessarily indicate a clinically relevant POPF (Grade B or C); rather, elevated drain amylase levels are needed for inclusion in our proposed algorithm. We added a comment on that in the 10th paragraph of our discussion, (Based on our study findings..)

Thank you for your nice comment regarding our proposed algorithm. Unfortunately, during the reviewing process, it was suggested by the editor to remove it, hence it is not available on the revised version.

We hope that with your comments, the manuscript is now more informative and explanatory

With regards,

the authors of the manuscript 

Round 2

Reviewer 3 Report

Comments and Suggestions for Authors

The authors correctly addressed all major concerns raised by the reviewers

Author Response

We sincerely thank the reviewer for their constructive feedback and for acknowledging that we have addressed the major concerns raised. We greatly appreciate your time and thoughtful review, which helped us improve the clarity and quality of our manuscript.

Reviewer 4 Report

Comments and Suggestions for Authors

I am very pleased to accept your article. You have explained the points I was curious about in a very elegant way—well done.

Author Response

We sincerely thank the reviewer for their positive feedback and kind words. We are glad that our explanations addressed your queries clearly, and we greatly appreciate your time and thoughtful evaluation of our work.